# Various AAV Serotypes and Their Applications in Gene Therapy: An Overview

**DOI:** 10.3390/cells12050785

**Published:** 2023-03-01

**Authors:** Shaza S. Issa, Alisa A. Shaimardanova, Valeriya V. Solovyeva, Albert A. Rizvanov

**Affiliations:** 1Department of Genetics and Biotechnology, St. Petersburg State University, 199034 St. Petersburg, Russia; 2Institute of Fundamental Medicine and Biology, Kazan Federal University, 420008 Kazan, Russia

**Keywords:** AAV, gene therapy approach, serotype, tropism, gene therapy medication

## Abstract

Despite scientific discoveries in the field of gene and cell therapy, some diseases still have no effective treatment. Advances in genetic engineering methods have enabled the development of effective gene therapy methods for various diseases based on adeno-associated viruses (AAVs). Today, many AAV-based gene therapy medications are being investigated in preclinical and clinical trials, and new ones are appearing on the market. In this article, we present a review of AAV discovery, properties, different serotypes, and tropism, and a following detailed explanation of their uses in gene therapy for disease of different organs and systems.

## 1. Introduction

Thanks to the rapid progress in biomedical technologies, large developments of novel high-tech treatment approaches have become possible today for several human diseases. However, despite scientific discoveries in the field of gene and cell therapy, some diseases still have no effective treatment. Advances in genetic engineering methods have enabled the development of effective gene therapy methods for various diseases based on adeno-associated viruses (AAVs). AAVs are effectively used to treat a number of genetic and acquired human diseases, more specifically, monogenic hereditary ones, i.e., diseases caused by mutations in one gene. Today, many AAV-based gene therapy medications are being investigated in preclinical and clinical trials, and new ones are appearing on the market for the treatment of numerous human diseases, including those that were previously considered incurable.

To date, there are only a few approved AAV-based gene therapy medications. For example, the US Food and Drug Administration (FDA) recently approved “Hemgenix”, a drug for the treatment of hemophilia type B, which is caused by congenital deficiency of factor IX. The drug is an AAV serotype 5 (AAV5), carrying the gene of deficient factor IX. As for hemophilia type A, Roctavian has also recently been approved by the FDA as an AAV5-based gene therapy vector, carrying the deficient factor VIII gene, driven by a liver-selective promoter [1]. Another breakthrough gene therapy medication is “Zolgensma”, which was intended for spinal muscular atrophy (SMA) treatment, a disease characterized by degeneration of the anterior horns motor neurons in the spinal cord resulting in a loss of their functions. Zolgensma is an AAV-serotype-9 (AAV9)-based vector encoding complementary DNA (cDNA) of the survival motor neuron 1 gene (survival motor neuron, SMN1). The drug was approved by the FDA in May 2019 for the treatment of type 1 SMA. Following intravenous administration, AAV9 crosses the blood–brain barrier (BBB) and provides targeted delivery of the SMN gene to neurons. It has been shown that intravenous administration of Zolgensma leads to an improvement in patients’ motor skills, along with a decrease in the clinical manifestations of SMA [2]. Another FDA-approved AAV-based gene therapy medication is Luxturna, which is intended for the treatment of retinal dystrophy caused by biallelic mutations in the *RPE65* gene (an enzyme of retinal cells involved in light-sensitive pigment regeneration). Drug delivery to the retinal cells leads to synthesis restoration of normal RPE65 protein, which is necessary for the regeneration of the photosensitive pigment [3]. The very first AAV-based gene therapy medication to be registered is Glybera, which was approved by the European Commission for the treatment of a rare hereditary disease, lipoprotein lipase deficiency [4].

Nevertheless, despite the successful development of AAV-based gene therapy medications, many preclinical and clinical studies still get suspended or terminated, without resulting in a drug registration, and therefore, many diseases, especially the rare ones, remain untreated. One explanation for that could be the lack of knowledge in gene therapeutics, in particular, features of AAV vectors. To tackle this issue, further research is needed to achieve better optimization and higher effectiveness of AAV-based gene therapy approaches. This review discusses various AAV serotypes, some features of infection, tissue and organ tropism, efficacy in many human diseases, etc.

## 2. Brief History of AAVs’ Discovery

AAVs are small non-enveloped DNA viruses belonging to the *Parvoviridae* family, that were first isolated in 1965, as a contaminant in preparations of a simian adenovirus (Ad) [5,6]. The viruses were found incompetent to productively infect cells without a co-infection by a helper virus, usually an Ad or any type of herpesviruses, and thereby they were named as “adeno-associated”, and classified into the Dependovirus genus [7,8]. After being regarded as defective viruses for a long time due to their codependency, later studies on AAVs disproved this theory and showed that they rather launch a latent infection in the host cell, that could convert to productive infection under stress [8,9]. Although AAVs have a high seroprevalence in humans (it has been estimated that 50% to 96% of the human population is seropositive for the second serotype of AAV (AAV2) depending on age and ethnic group) [10,11,12], they, however, were not linked to any disease neither in humans, nor in any other species [8]. Different AAVs have not only been detected in primates isolates [13], but also from avian [14,15,16], caprine [17,18], bovine [19,20], and equine stocks [21].

## 3. Properties, Structure, and Genome Organization of AAVs

Aside from AAV5 being the most divergent, all AAVs share a similar structure and properties [22]. AAVs are easy to manipulate, as their particles can maintain biological stability in extreme conditions of pH and temperature [23,24]. They share a genome of approximately 4.7 kb single-stranded DNA packed into an icosahedral, non-enveloped capsid with a diameter of 20–25 nm [6,25]. The AAV genome consists mainly of two viral genes: rep (replication) and cap (capsid), flanked by inverted terminal repeats (ITRs) [12,25]. As the ITRs have a palindromic nucleotide sequence, they create characteristic T-shaped hairpin structures, providing essential structural elements for viral genome replication and packaging [26]. ITRs also play a regulatory role in viral gene expression and host genome integration [10,27]. The open reading frame (ORF) of rep encodes several nonstructural proteins that are required for gene regulation, replication, transcription, and encapsidation [12,28,29], while the ORF of cap encodes three structural proteins: virion protein 1 (VP1), VP2, and VP3, with a molar ratio of 1:1:10 in AAV particles [10,30,31]. Distinct tissue tropism of different AAV serotypes results from variations in the processing of this cap ORF, leading to variant immune and transduction profiles [12,32].

## 4. AAV Serotypes and Tropism

Depending on their serotype, AAVs can have specific tropism for specific organs and tissues of the body. There are different AAV serotypes that vary in many aspects. Next, each serotype will be separately discussed. Table 1 summarizes the characteristics and properties of AAV serotypes, and Figure 1 demonstrates their variant tropisms.

### 4.1. AAV1

The exact origin of the first serotype of AAV (AAV1) is still unknown, as it was not initially isolated from tissues, but as a contaminant of Ad stocks [33,34], and its antibodies were found both in humans and non-human primates (NHPs) [35]. This serotype uses sialic acid as its primary cellular surface receptor [36], and AAV receptor (AAVR) as a coreceptor [37]. According to Rabinowitz J.E. et al., AAV1 does not bind heparin as it lacks R585 and R588, the amino acid residues required for such binding, and thereby it cannot be purified using heparin [33,38]. Zolotukhin S. et al., developed a protocol for AAV1 chromatographic purification by iodixanol gradient centrifugation and anion-exchange chromatography [39]. It can also be purified using mucin columns, as it can bind the sialic acid residues in mucin [40]. Moreover, recombinant AAV1 (rAAV1) was not found to contain any detectable post-translational modifications (PTMs), according to a systematic analysis conducted on ten AAV serotypes by Mary B. et al. [41], and it was the first viral vector to be approved for use in gene therapy [42]. In 1999, Xiao W. et al., conducted a study to investigate viral vectors for gene therapy, as a result of which, AAV1 was found to be the most efficient serotype for skeletal muscles’ transduction [35]. Many studies have since confirmed AAV1 high tropism to skeletal muscles of murine, canine, and NHP origins, compared to other serotypes [43,44,45,46]. AAV1 was also found to achieve an efficient transduction of neurons and glial and ependymal cells in the murine brain [47]. Moreover, it was found to be able to effectively transduce the heart [48,49], endothelial and vascular smooth muscles [50], and retina [51].

### 4.2. AAV2

AAV2 is considered the most studied serotype among all AAVs [45]. It was first discovered in 1965 as a contaminant of simian Ad preparations [52]. Later, in 1998, its primary cellular receptor, heparan sulfate proteoglycan (HSPG), was identified by Summerford C. and Samulski R.J. [53], and the amino acid residues providing its affinity to HSPG were suggested, afterwards, as R585 and R588 [54]. Accordingly, rAAV2 can be purified using heparin column affinity chromatography [55]. Nevertheless, the binding of AAV2 to its primary receptor was found to be insufficient for cell entry, so several coreceptors were later identified for it [52], including the human fibroblast growth factor receptor 1 (FGFR1) [56], αVβ5 [57] and α5β1 integrins [58], hepatocyte growth factor receptor (HGFR) [59], laminin receptor (LR) [60], and CD9 [61]. The capsid of rAAV2 is reported to acquire multiple PTMs, including ubiquitination, phosphorylation, SUMOylation, and multiple-site-glycosylation [41]. As the most studied serotype, AAV2, in fact, has shown various tropisms for several tissues in NHP, murine, canine, avian, and human cell types, including renal tissue [62,63,64], hepatocytes [65,66], retina [67,68,69], non-mitotic cells of the central nervous system (CNS) [70,71], and skeletal muscles [45,71]. Nonetheless, broader tissue tropisms of AAV2 were enabled by the further innovation of mosaicism and cross-packaging (or cross-typing) of AAVs (explained in detail later in this article), where the viral genome of one serotype could be packaged into a capsid of another type, providing a wider transduction spectrum [38,72,73]

### 4.3. AAV3

The third serotype of AAV (AAV3) was originally isolated from humans [34,45]. Similar to AAV2, this serotype uses HSPG, FGFR1, and LR receptors [60,74], along with the human HGFR (hHGFR) receptor [60]. Iodixanol gradient ultracentrifugation along with ion exchange chromatography have been used for AAV3 purification [75]. PTMs of rAAV3 capsid include acetylation, phosphorylation, and glycosylation [41]. Due to its inadequate transduction efficiency in vitro and in murine cell lines, AAV3 was mostly overlooked as a choice for gene therapy [76]. However, as it has been later found to use hHGFR as a coreceptor, it showed extremely efficient transduction of human liver cancer cells as well as human and NHP hepatocytes [76,77]. Since this selective tropism of AAV3 has been discovered, various studies have aimed at optimizing strategies to generate rAAV3 vectors with a higher transduction efficiency [76,78,79,80,81]. The developed strategies suggested different approaches, mainly, capsid modification of AAV3 vectors [76,80,81] and modification of hHGFR expression levels, along with the tyrosine kinase activity associated with it [78]. AAV3 was also found to have specific tropism to cochlear inner hair cells, showing high in vivo transduction efficiency in a murine model [82].

### 4.4. AAV4

The fourth serotype of AAV (AAV4) is considered one of the most antigenically distinct serotypes [83]. It has reportedly originated in NHPs [84], mainly in green African monkeys [85], as antibodies to its viral particles have been detected in their sera [86,87]. A study of the AAV4 structure showed that its capsid surface topology shares a significant similarity with that of human parvovirus B19 and Aleutian mink disease virus [83]. AAV4 uses a-2,3-O-linked sialic acid for cell binding and infection [88]; accordingly, mucin columns can be used for AAV4 purification, based on its ability to bind sialic acid residues in mucin [33,40]. In addition, as this serotype lacks heparin-binding activity, it cannot be purified using heparin column affinity chromatography like AAV2, however, ion-exchange chromatography procedures have been developed and proven a high purification efficiency [89]. The only reported PTM of rAAV4 is ubiquitination of its capsid proteins [41]. AAV4 is suggested to be able to transduce human/NHP cells, as well as cells of murine and canine origins [45,87]. The specific tropism of AAV4 results in a transduction efficacy of specific cell types in the mammalian central nervous system (CNS), mainly the ependymal cells [90]. Moreover, following subretinal delivery, AAV4 has showed stable transduction of retinal pigmented epithelium (RPE) cells of rodent, canine, and nonhuman primate models, a distinctive feature enabled by the specificity of its capsid [91]. In a murine model, AAV4 has also shown a significant transduction rate of kidney, lung, and heart cells [92].

### 4.5. AAV5

As it was first isolated in 1983 from male genital lesions, AAV5 became the only AAV serotype to be isolated directly from a human tissue [93]. This serotype is considered the most genetically divergent of all AAVs [6,94,95,96], with a variety of unique characteristics, such as the distinct size and function of its ITR regions [22], utilizing herpes simplex virus (HSV) as its helper virus for human infections [94], and using an atypical endocytic route as a pathway for viral entry [33]. Another distinctive feature of AAV5 is its ability to transduce cells that cannot be transduced with AAV2, an exclusive advantage for gene therapy uses [97,98,99]. AAV5 was also found to use sialic acid as its primary receptor [88,100,101], along with platelet-derived growth factor receptors (PDGFR) α and β as coreceptors [102,103]. Similar to AAV1 and AAV4, mucin columns can be used for AAV5 purification [104,105], and ion-exchange chromatography procedures have also been developed for that aim [89]. Capsid proteins of rAAV5 are reported to undergo multiple PTMs, including ubiquitination, phosphorylation, SUMOylation, and glycosylation [41]. AAV5 proved to have a significant transduction efficiency for murine retinal cells [38,51], mainly for photoreceptor cells [106]. In addition, AAV5 tropism has been investigated in the murine brain, and proved, as a result, a transduction competence for multiple neural cell types, including Purkinje cells, stellate, basket, and Golgi neurons, and it was able to reach to the inferior colliculus and ventricular epithelium [107,108]. AAV5 is also known for its efficient transduction of murine airway epithelia by apical infection [98,109,110,111], vascular endothelial cells, and smooth muscles [50]. It is also reported to have tropism for liver cells in mice [92,112].

### 4.6. AAV6

The classification of the sixth serotype of AAV (AAV6) is still a matter of controversy, as it presents high genomic similarity with both AAV1 and AAV2 serotypes, however, it has still been assigned its own serotype numbering [33,35]. AAV6 has a serological profile almost identical to that of AAV1, and shares its sequence of coding region with a homology percentage of 99%, along with multiple regions identical to those of AAV2 [113]. Accordingly, it was suggested to be a naturally occurring hybrid resulting from homologous RECOMation between AAV1 and AAV2 [35,45,113,114]. AAV6 was first isolated from a human Ad preparation [45,113,115], and similar to AAV1, was found to bind sialylated proteoglycans, mainly α2,3-/α2,6-linked sialic acid, as its primary receptor, as well as binding heparan sulfate [36,116,117]. As for its coreceptor, it binds epidermal growth factor receptor (EGFR) [118]. The only reported PTM of rAAV6 is acetylation of its capsid proteins [41]. Similar to the previously described serotypes, AAV6 can be purified by either heparin or mucin column affinity chromatography, as it can bind both [40]. AAV6 is reported to have tropism for a variety of tissues, including airway epithelia of murine and canine models [119,120], murine liver cells [92,121], and skeletal muscles of murine and canine models, with a transduction efficiency even higher than that of AAV2 [92,122,123,124], cardiomyocytes in murine [92,125], porcine [126], canine [127], and in sheep [128] models.

### 4.7. AAV7

The seventh serotype of AAV (AAV7) was first isolated in 2002 from NHP tissues, specifically, from Rhesus macaque monkeys [35,129]. Its mechanisms of cell binding and cell entry are still unknown [52,130], but it is established that this serotype does not bind heparin, or any other glycan in general [131]. Capsid proteins of rAAV7 undergo multiple PTMs, including glycosylation, primarily, along with phosphorylation, SUMOylation, and acetylation [41]. A study by Calcedo R et al., investigated the epidemiology of AAV-neutralizing antibodies in the worldwide population, and found that the seroprevalence of AAV7 antibodies is relatively low in humans, an advantage of this serotype to be used in clinical applications [132]. Viral vectors based on AAV7 proved a high efficiency of transduction for skeletal muscle cells in murine models, similar to that achieved by AAV1, and higher than AAV2 [133]. This serotype also proved a strong tropism to hepatocytes in murine [92] and human [92] tissues. In CNS of NHPs, AAV7 viral vectors were found to achieve a robust transduction mainly in cortical and spinal tissues [134]. Moreover, AAV7-based viral vectors can, reportedly, achieve a significantly high transduction rate of murine neurons [135] and photoreceptor cells in the retina both in vivo and ex vivo [136]. AAV7 vectors also appear to have a limited tropism to vascular endothelial cells, that could be relatively enhanced through proteasome inhibition [130], and an in vivo transduction preference for epicardium cells in murine cardiac tissue [49].

### 4.8. AAV8

Similar to AAV7, the eighth serotype of AAV (AAV8) was first isolated in 2002 from Rhesus macaque monkeys [35,129]. As a primary receptor, AAV8 binds LR, the same receptor used by AAV2 and AAV3 [45,60]. Various procedures for rapid and scalable purification of AAV8 have been developed since its discovery, including, for example, dual-ion-exchange chromatography [137,138], or iodixanol gradient centrifugation [139,140]. Phosphorylation, glycosylation, and acetylation are the three PTMs reported for rAAV8 capsid proteins [41]. AAV8 is best known for its strong tropism to liver cells, and accordingly, its transduction efficiency of hepatocytes, which is far stronger and faster than those of all other AAV serotypes in different models, including murine, canine, and NHP [92,112,141,142,143,144,145,146,147]. Following systemic delivery in murine models, AAV8 was proven to be the most efficient serotype for transduction of both skeletal and cardiac muscles, owing to its ability to cross the blood vessel barrier, a feature that both AAV1 and AAV6 lack, limiting their efficiency of muscle transduction to local delivery only [148]. AAV8 could also achieve a successful in vivo transduction of murine pancreatic cells, following localized delivery [149,150]. Moreover, a high-rate transduction of murine renal cells could be reached by local direct delivery of AAV8 viral vectors into the kidney tissue [151]. AAV8 was also found to achieve an efficient transduction of different cells in the murine retina, including amacrine, Müller, and putative bipolar cells, along with some horizontal cells and cells in the ganglion cell layer (GCL) [152,153,154]. AAV8 transduction efficiency appears to be susceptible to proteasome levels in some tissues, and accordingly, could be increased using proteasome inhibitors [130].

### 4.9. AAV9

The ninth serotype of AAV (AAV9) was first identified in a human isolate in 2004, and was named a new serotype as it had a serological profile distinct from the previously known AAVs, however, it was suggested to be closely related to clades containing AAV7 and AAV8 [129,155]. As a primary receptor, AAV9 uses terminal N-linked galactose [156,157], and it is also suggested to bind a putative integrin, along with LR as coreceptors [60,156]. Scalable simple purification protocols have been developed for AAV9 purification, including ion-exchange chromatography [24] and sucrose gradient centrifugation [158]. Capsid of rAAV9 has one of the highest totals of PTMs, including multiple ubiquitination, phosphorylation, SUMOylation, and glycosylation modifications, along with acetylation [41]. In most tissues, AAV9 seems to achieve cell transduction with efficiency superior to other AAVs [33,129]. For example, in a study aimed to investigate AAV1–9 distribution following systemic delivery in a murine model, AAV9 has shown rapid-onset, the best genome distribution, and the highest protein levels, in comparison with all other AAVs [92]. Moreover, in the CNS of murine, NHP, and feline models, it has a unique feature compared to other serotypes, as it can cross the BBB and transduce not only neuronal but also non-neuronal cells, including astrocytes, that cannot usually be transduced by other AAVs [27,159,160,161,162], also showing tropism to photoreceptor cells in the retina [163]. Viral vectors based on AAV9 have also proven to be more efficient than those of AAV1 and AAV8 (in some cases 5–10-fold higher than AAV8), for murine, NHP, and porcine cardiac muscle transduction [164,165,166,167,168,169], enabled with another distinctive feature of AAV9—its ability to traverse the physical barrier of vascular system endothelia [168]. Another example of AAV9 superiority over other AAVs was presented in a study by Inagaki K et al., where the serotype achieved robust transduction of murine hepatocytes, skeletal muscles, and pancreatic cells [81,169]. AAV9-based viral vectors also appear to be tropic to murine photoreceptor cells [170], renal tubular epithelium cells [171,172,173], Leydig cells in the testicular interstitial tissue [174], and alveolar and nasal epithelia [175,176].

### 4.10. AAV10 and AAV11

The ninth and tenth serotypes of AAV (AAV10 and AAV11) were first found and described in 2004 in NHP isolates, namely from cynomolgus monkeys, with capsid proteins of great resemblance to AAV8 and AAV4, respectively [34,177], resulting in serological cross-reactivity with those two serotypes [178]. However, antisera against AAV10 and AAV11 were not found to have any cross-reactivity against those of AAV2, which recommended them as good viral vector candidates for gene therapy in individuals having antibodies against the latter [33]. It remains unknown what cellular receptors and coreceptors AAV10 and AAV11 use for cell binding and entry [37,115]; therefore, procedures describing their purification protocols are generally based on iodixanol gradient centrifugation [179]. Similar to AAV9, capsid of rAAV10 has one of the highest totals of PTMs, including mainly multiple glycosylation and phosphorylation modifications, along with ubiquitination, SUMOylation, and acetylation [41]. A study investigating biological distribution of both AAV10 and AAV11 in monkeys suggested them to be tropic to NHP intestinal cells, hepatocytes, lymph nodes, and less frequently, to renal cells and adrenal glands [180]. AAV10 was also suggested to have tropism to murine small intestine and colon cells [181]. Compared to AAV8 and AAV9, AAV10 appears to have the largest tropism range to murine retinal cells, as it can reportedly transduce a variety of cell types, including RPE, cells in the ganglion cell layer, several cell types in the inner nuclear layer, photoreceptors, and a highly efficient transduction of horizontal cells [152]. Following intravenous delivery, AAV10 was found to target murine liver and lung cells [182], however, upon localized delivery, it transduced murine renal and pancreatic cells [183]. As for AAV11, it was found to have mild tropism to NHP CNS, mainly to the cerebrum and spinal cord [180]. A recent study of neural gene therapy using rAAV11 found that murine projection neurons and astrocytes could also be targeted using this serotype [184].

### 4.11. AAV12

The twelfth serotype of AAV (AAV12) was first isolated from a simian Ad stock, and then characterized as a novel serotype, as it exhibited distinctive biological and serological properties [185]. Although it was proven that AAV12 does not use heparan sulfate proteoglycans or sialic acids for attachment and cell entry, it remains unknown how exactly it binds target cells [177,185]. However, according to a study investigating components of a potential receptor complex for AAV12, mannose and mannosamine were suggested as components of such complex, as they inhibited AAV12 cell transduction [186]. Moreover, being resistant to neutralization by human antibodies, AAV12 represents a good candidate for human gene therapy applications [185,187]. AVB Sepharose affinity chromatography has been used for purification of AAV12 [188]. In murine models, it has shown tropism to salivary glands and muscles [185]. It has also shown strong, localized in vivo tropism to murine nasal epithelia following intranasal administration [187].

### 4.12. AAV13

The thirteenth serotype of AAV (AAV13) is another simian Ad that appears to bind HSPG, although its primary cell receptor remains unknown [189,190,191]. It was also found to share structural similarity with AAV2 and AAV3, making it the closest related AAV to those two serotypes, with a capsid conserving all AAV capsids’ structural features [188], but there are only limited data on this serotype tropism and transduction efficiency [191].

### 4.13. Novel Hybrid AAV Vectors

In addition to the natural AAV serotypes described above, novel AAV vectors have been developed during the last two decades, and are still being developed [192]. Using different engineering strategies, novel hybrid vectors have been generated in order to enhance their transduction, modulate their immunogenicity, or limit their tropism to specific cells [193,194,195,196,197]. There has been different types of such engineered novel vectors, including mosaic, chimeric, and combinatorial vector libraries [33]. Mosaic vectors have multiple subunits of various serotypes in their capsid, chosen according to their properties of receptor binding and intracellular trafficking [33,198]. In chimeric virions, the capsid usually has modified protein generated by domain swapping and DNA shuffling strategies to alter specific amino acids [33,194]. Combinatorial vector libraries also use DNA shuffling and error-prone PCR methods to generate AAV libraries of novel serotypes with mixed genomes [199,200].

## 5. AAV as Viral Vectors for Gene Therapy Applications

A variety of AAV features have made it an appealing viral vector candidate to be used in gene therapy applications. To begin with, AAVs are non-pathogenic viruses, since no diseases have been linked to them [201]. AAV vectors have also proven their ability to provide stable integration into the target genome [202,203,204,205], and such feature is highly needed for gene therapeutic uses considering the high level of expression that could be maintained as a result [206], without affecting the target cell functions in the long term [207,208,209]. Moreover, having only ITRs, AAV vectors cannot interfere with the inserted gene regulation either [201]. More specifically, a unique feature of AAV vectors, contrary to all currently available gene-editing platforms, is utilizing the homologous recombination pathway, which does not involve exogenous nucleases, providing, therefore, a highly accurate editing process that preserves genome integrity without adding to the mutational burden of the target site [210]. Thanks to such unique gene-editing properties, AAV vectors are currently the leading platform for in vivo gene therapy delivery [210,211]. Another attractive feature is the very broad range of cells, tissues, and hosts that AAV can efficiently transduce in vivo and in vitro [87,201], including dividing and non-dividing cells [212,213] in humans [99], and NHP [214,215], murine [216,217,218], canine [219,220,221], feline [222,223], and a variety of other models. In addition to the genetic properties, AAVs’ distinct physical properties represent more reasons why they should be used for gene therapy, such as easy manipulation, resistance to pH changes, heat changes, and detergents [87]. Next, we present various studies that investigated AAV vectors for a wide range of gene therapy applications (reviewed in Table 2).

### 5.1. AAV Viral Vectors for Gene Therapy of the CNS

AAV-mediated CNS gene therapy was first believed to predominantly target neurons, with lesser chance to affect other cells and tissues in the CNS [224], a fact that was later proven to be inaccurate in many studies [134,160,214,225]. For a variety of neurodevelopmental and neurodegenerative diseases, AAV-mediated gene therapy has been tested using different administration routes, including intraparenchymal, intrathecal, intracerebroventricular, and intracisternal injection, many of which have shown promising results [226].

Parkinson’s disease (PD) has been one of the most studied neurological disorders as a target for AAV-mediated gene therapy [224,227,228]. In a phase I clinical trial, Kaplitt et al., investigated an AAV-mediated gene therapy approach for advanced PD patients, where serotype 2 was used as a vector for unilateral subthalamic delivery of the glutamic acid decarboxylase (*GAD*) gene [229]. Providing a significant improvement in motor function scores up to 12 months after surgery, the approach was found to be safe and well-tolerated in patients. A similar double-blind, controlled, randomized clinical trial conducted later by LeWitt et al., investigated the same AAV2-*GAD* vector for bilateral subthalamic delivery in advanced PD patients, and showed similar results of safety and improvement of motor function [230]. Bartus et al., also tested the bilateral stereotactic delivery of AAV2-neurturin in PD patients of an open-label clinical trial, the initial obtained data of which supported the feasibility, safety, and good tolerance of the approach as a potential treatment for PD [231]. However, bilateral intra-striatal infusion of an AAV2 vector containing the aromatic L-amino acid decarboxylase (*AADC*) gene in moderately advanced PD patients led to an improvement in PD rating scales that was associated with a risk of intracranial hemorrhage in patients, along with headaches [232]. As for sustainable transgene expression, a clinical trial conducted by Mittermeyer G. et al., investigated the potentials of rAAV2 carrying the aromatic L-amino acid decarboxylase gene (rAAV2-*AADC*) [233]. In the trial, 10 patients with moderately advanced PD received bilateral infusions of recombinant vector into the putamin, showing good tolerance and a stable expression of transgene that lasted for the following 4 years, although higher vector doses were suggested for further studies. In children with AADC deficiency, the same delivery route for the same vector (rAAV2-*AADC*) was also assessed in an open-label phase I/II clinical trial [234]. The therapy was well-tolerated in general, providing evidence for potential improvement of motor function.

Similarly, spinal muscular atrophy (SMA) has always represented an attractive candidate for gene therapy, being caused by a single gene defect that affects the survival motor neuron (SMN) protein [235]. Therefore, SMA has been suggested as a target for AAV-mediated therapeutic approaches, mainly using AAV9 [161]. An open-label, phase I clinical trial was designed by AveXis, Inc. in 2014 to assess the safety and efficacy of AA9-*SMN* as a treatment for SMA [236]. Following single-dose intravenous administration of the vector, a significant improvement of the motor function in all 15 patients was observed, reflected by their ability to perform different activities, such as unassisted sitting and walking, oral feeding, and speaking, with no reported motor function regression at the two-year follow-up [236]. A long-term safety assessment, however, was recommended.

Lysosomal storage diseases (LSDs), another group of neurodegenerative diseases, have also been extensively targeted by AAV-mediated gene therapy [224]. One example of which is mucopolysaccharidosis type VII (MPS VII or Sly disease), a disease that results from genetic deficiency of beta-glucuronidase (GUSβ) [237]. In the murine model, AAV-mediated gene therapy has been shown to provide stable expression of the deficient enzyme upon single administration, that was adequate for phenotype correction in the liver and most of the neuraxis [238,239]. As for neonatal murine model, intravenous delivery of rAAV-*GUSβ* proved to yield therapeutic levels of the deficient enzyme in multiple organs, including ones of the CNS, with the gene expression not being affected by rapid growth and differentiation of tissues [240]. The canine model of MPS VII has also shown promising therapy results following rAAV-GUSβ intrathecal injection of serotypes 9 and the NHP-derivate AAVrh10 [221], as high expression levels of GUSβ were detected in the CNS tissues, with the enzyme in brain tissue homogenates showing over 100% normal activity. Another type of MPS, MPS IIIA, has also been assessed for gene therapy in the canine model using AAV9 carrying the deficient enzyme’s gene [241]. Following intra-cerebrospinal fluid (CSF) delivery, the administered vector provided sustained and widely distributed transgene expression with no toxicity for a duration of seven years after therapy. Metachromatic leukodystrophy (MLD) is also one of the LSDs that results from arylsulfatase A (ARSA) enzyme deficiency and has been targeted by AAV-mediated therapeutic approaches [242]. In the murine model of MLD, as well as in NHPs, serotypes 1, 5, 9, and rh10 carrying the *ARSA* gene have provided supporting results of their therapeutic potentials as gene therapy viral vectors [243,244,245,246]. GM2 gangliosidoses is another example of lysosomal storage diseases that have been assessed for AAV-mediated gene therapy, including Tay-Sachs disease (TSD) and Sandhoff disease (SD) [247,248,249]. Preclinical data from many in vivo studies have provided evidence of AAVs’ efficacy for GM2 gangliosidoses in murine [250], ovine [251], and feline [252] models. Clinically, there has been trials applying AAV-mediated gene therapy for different LSDs throughout the last decades with variant outcomes [242,248,253]. In 2018, four children with asymptomatic or early-stage MLD underwent a phase I/II trial for gene therapy using AAVrh.10-h*ARSA* [253]. Although there was a significant elevation in ARSA levels recorded in the CSF following intracerebral vector delivery, no clinical improvement was noticed compared to the control group. As for gangliosidosis, the first clinical trial for gene therapy of Tay-Sachs disease using an AAV vector was conducted by Taghian et al., in 2020 [248]. The treatment proved to be safe for the two treated children and resulted in a broad distribution of the transgene in CNS.

Canavan disease (CD) is another rare inherited leukodystrophy belonging to LSDs, that has been targeted by AAV-mediated gene therapy [254]. CD results from aspartoacylase (ASPA) enzyme deficiency, that leads to N-acetylaspartic acid (NAA) accumulation in the brain, causing white matter degeneration [255]. In a murine model, rAAV2-*ASPA* delivery into the striatum and thalamus of the brain resulted in an increased ASPA activity and, thereby, decreased NAA accumulation and white matter degeneration. However, areas remote from the injection site, such as the cerebellum, were not affected [255,256]. In 2002, Janson et al., suggested a clinical protocol using rAAV2-*ASPA* for gene therapy in CD patients [257], and currently there are several ongoing clinical trials testing different serotypes of rAAV (including 2, 9, and olig001), carrying the *ASPA* gene for CD gene therapy [258,259,260].

Krabbe disease, also known as globoid cell leukodystrophy (GLD), causes demyelination in a similar way. The diseases results from deficiency of the lysosomal enzyme galactocerebrosidase (GALC), leading to accumulation of its substrate, psychosine, which affects both the central and peripheral nervous systems [261]. During the last two decades, a large number of in vivo studies have been conducted to investigate different AAV serotypes as viral vectors for gene therapy of GLD in different animal models, including murine and canine. Intracerebral delivery of rAAV1-*GALC* has been tested in the murine model of GLD in an in vivo study conducted by Rafi et al. [262]. The approach resulted in sustained expression of the deficient enzyme, and thereby positively affected myelination status, and animals’ lifespan. However, both treated and untreated mice died with similar symptoms, suggesting that the used approach should be initiated prior to symptoms’ onset. rAAV2/5-*GALC* has also shown therapeutic efficacy in the murine model of GLD following intracranial delivery [263,264]. In the study conducted by Lin et al., in 2011, mice treated with rAAV2/5-*GALC* showed wide dispersion of the GALC transgene across the CNS, reaching areas remote from the injection site [263]. Moreover, rAAV2/5-*GALC* delivery resulted in a reduced loss of oligodendrocytes and Purkinje cells, along with a significant improvement of neuromotor function and a prolonged lifespan of treated mice. Karumuthil-Melethil et al., conducted another preclinical study using recombinant vectors AAV9, AAVrh10, and AAV-Olig001, carrying the *GALC* gene, for GLD gene therapy in the murine model, following lumbar intrathecal delivery [265]. All three serotypes provided wide distribution of the transgene across the CNS and liver, resulting in a significant improvement of myelination, and a prolonged lifespan, with AAV9 being the most effective when combined with bone marrow transplantation. AAV9 and AAVrh10 have also been tested for GLD therapy in the canine model [266,267]. Combined intravenous and intracerebroventricular delivery of AAVrh10 encoding canine *GALC* (AAVrh10-*cGALC*) resulted in delayed symptoms onset, prolonged lifespan, correction of biochemical defects, and also positively affected neuropathology in treated animals [266]. Similarly, AAV9-*cGALC* has shown promising therapeutic potency in the GLD canine model [267]. Reportedly, intrathecal delivery of AAV9-*cGALC* resulted in increased activity of the deficient enzyme and, therefore, normal levels of its substrate. It also improved myelination, and decreased inflammation both in the CNS and PNS, which, along with prevention of clinical neurological dysfunction, resulted in a significantly prolonged lifespan of treated dogs, compared to the control group. However, sufficient dosing was found to be critical, as high doses significantly extended the lifespan even for post-symptomatic subjects, and a 5-fold lower dose of the vector resulted in an attenuated form of disease [267].

For ocular diseases, AAV viral vectors have also provided great therapeutic opportunities both in vivo and in human clinical trials [136,170,268,269,270,271]. In a study by Petrs-Silva et al., the efficacy of intraocular transduction of rAAV serotypes 2, 8, and 9 was investigated in mice for targeting ocular neurovascular and retinal diseases [170]. Besides transduction of retinal cells, both serotypes 8 and 9 efficiently transduced the ganglion cell layer, providing evidence for their potentials in therapeutic approaches of ocular diseases. Moreover, AAV2/7 and AAV2/8 hybrid serotypes have achieved high transduction rates of the murine photoreceptors, showing promising potentials for treatment of inherited photoreceptor diseases [136]. AAV2/6 has also shown an efficient transduction of murine cone photoreceptors following subretinal injection [268]. In humans, AAV2/4 has been used in a clinical trial for treatment of childhood blindness caused by Leber Congenital Amaurosis (LCA) disease, mainly by a mutation in the retinal pigment epithelium-specific protein (RPE65) [269,270]. Over a follow-up period between 1 and 3 years, subretinal injection of AAV2/4-*RPE65* showed good tolerance both systemically and locally, with no adverse effects, and resulted in an improvement of visual function presented by different parameters in different patients, including improvement of visual acuity and color vision, as well as a reduction of visual fatigue or photophobia [269]. Another example of AAVs’ applications in ocular disease therapy is a phase I/II clinical trial by MacLaren R et al., that has aimed to treat blindness caused by choroideremia, an X-linked recessive disease that results from a mutation affecting retinal escort protein 1 (REP1) [272,273]. In the trial, AAV2-*REP1* resulted in improvement of retinal sensitivity in all six patients, following subretinal injection, with two of them having significant increases in visual acuity, supporting further consideration of the tested therapeutic approach [272]. Various other clinical trials have applied AAV-mediated gene therapy for ocular diseases, and many of which have presented promising results, using mainly AAV2 and AAV8 [274,275,276,277,278,279,280,281] (detailed in Table 2).

Similar to ocular diseases, hearing disorders have also been suggested for AAV-mediated gene therapy [282]. In a porcine model, Lalwani A. et al., tested AAV9 intracochlear delivery to assess its efficacy as a vector for gene therapy of hearing disorders, and demonstrated, as a result, transgene expression in the inner ear of animals following administration [282].

### 5.2. AAV Viral Vectors for Gene Therapy of Respiratory Diseases

For over 20 years, AAV vectors have been vastly investigated for gene therapeutics of respiratory diseases, both in preclinical experiments and human clinical trials [283,284]. However, a key limitation was that many AAV serotypes cannot efficiently transduce airway epithelial cells through the apical surface, suggesting molecular modifications of such serotypes to enhance their transduction efficiency [196]. Cystic fibrosis (CF) represents one of the most studied diseases as a target for AAV-mediated gene therapy [285]. In rabbits, AAV has proven to promote an efficient and stable gene transfer of the cystic fibrosis transmembrane conductance regulator (*CFTR*) gene into the airway epithelium, indicating, as a result, the vector potential to be used for gene therapy [286]. Subsequently, since 1998, clinical trials have started using recombinant AAV viral vectors to target different sites of the airway epithelium of CF patients, using variant administration routes [285,287,288]. The very first clinical trial to do so used the maxillary sinuses for delivery of recombinant AAV2 containing the *CFTR* gene (AAV2-*CFTR*) in ten CF patients, demonstrating a safe, successful transduction of targeted cells and a detected function restoration of the sinuses [288]. Soon after, a following phase II double-blind clinical trial tested unilateral administration of the same vector into the sinus for 23 CF patients, with an in-patient control, achieved by administering a placebo drug into the other sinus [289]. The approach again showed safety and good tolerance, although it did not confirm clinical effectiveness of the treatment. Aitken et al., also tested AAV2-*CFTR* in a phase I clinical trial for twelve mild CF patients using aerolization by nebulation for delivery, confirming the approaches safety, but failing to yield an effective clinical treatment [290]. Further trials on AAV2-*CFTR* with single or repeated dosing have yielded similar results of safety and good tolerance, providing little evidence of clinical treatment after intranasal and endobronchial delivery [291,292,293]. In order to optimize such therapeutic approaches and produce a functional CFTR in CF patients, further modifications of recombinant AAV vectors have since been developed [294,295]. Alpha-1 antitrypsin (α1AT or AAT) deficiency is another lung and liver disease that has been extensively studied as a target for AAV-mediated gene therapy both in vivo and in humans [284,296]. Reportedly, intravenous delivery of recombinant AAV vector carrying the human alpha-1 antitrypsin gene (AAV-h*AAT*) in the murine model resulted in potentially therapeutic serum levels of AAT [297,298]. Further research found that intrapleural delivery of rAAV2-h*AAT* or rAAV5-h*AAT* could achieve higher AAT levels both in lungs and serum compared to intramuscular delivery in C57BL/6 mice, with the rAAV5 showing 10-fold higher effectiveness than rAAV2 [299,300]. However, rAAV6/2-h*AAT* has been shown to transduce murine lung cells even more efficiently than rAAV5 both in vivo (in murine lung cells) and in vitro (in human airway epithelial cell culture) [301]. Similarly, rAAV8-h*AAT* has also been shown to provide murine lung cells’ transduction superior to that of rAAV5 following intratracheal delivery, as it resulted in 6-fold and 2.5-fold AAT levels in serum and broncho-alveolar fluid, respectively [302], and a high transduction rate of murine hepatocytes [298]. Chiuchiolo M et al., also proved the safety of rhAAV10-h*AAT* as a viral vector for treatment of AAT in wild-type murine and NHP models [303]. Besides, the delivered vector resulted in persistent expression of the transgene in chest cavity cells of both models, suggesting the efficacy of the tested therapeutic approach [303,304]. Examples of other rAAV vectors’ therapeutic potentials for AAT treatment have been shown in a variety of other preclinical studies, including rAAV1-AAT, rAAV2/9-AAT, and rAAV6 [46,120,175,305].

AAVs have also been tested to develop therapeutic approaches for other lung diseases, such as asthma and surfactant B deficiency [306,307]. A preclinical study by Zavorotinskaya et al., investigated the efficacy of rAAV carrying the interleukin 4 gene (*IL-4*) as a vector for gene therapy of allergic asthma, upon intratracheal delivery into the murine model [306]. Showing a significant inhibition of airway eosinophilia and mucus production along with a reduction in airway hyper-responsiveness and asthma-associated cytokines levels, obtained data suggested rAAV efficacy for gene therapy of the studied disease. Kang et al., have recently tested an engineered rAAV6/2 vector carrying human or murine surfactant protein B gene (*SFTPB*) in a murine model [307]. Mutations of this gene cause surfactant protein B deficiency (SPB) in humans, a rare genetic condition with a very poor prognosis [284,308]. In their study, Kang et al., demonstrated efficient transduction of the airway and alveolar epithelium by engineered rAAV6/2-*SFTPB* following intratracheal administration into a murine model of SPB. As a result, the administered vector was well-tolerated with no adverse effects. Moreover, rapid, long-term restoration of the deficient SPB protein was reported, along with an improvement of lung function, leading, subsequently, to an extended survival [307].

### 5.3. AAV Viral Vectors for Gene Therapy of Muscle Diseases

Having a large, body-distributed mass, and myofibers with a long half-life time makes muscles an attractive target for gene therapy, also considering the minimal invasiveness of the intramuscular delivery route [133,309]. Muscular dystrophies (MD) constitute a large group of muscle diseases that have been studied for a long time and proved to be a good target for AAV-mediated gene therapy [310]. For example, AAV1 carrying the follistatin gene, an antagonist of muscle growth negative regulator, has been shown to promote sustained improvement in muscle size and strength in NHPs following intramuscular administration [311]. There has also been many reports of both histopathological and functional sustained muscle restoration as a result of using AAV-mediated therapeutic approaches in animal models of Duchenne muscular dystrophy (DMD) [312,313], and Limb-Girdle muscular dystrophies (LGMD) [314,315,316,317,318]. As for clinical research, there have been several registered clinical trials of AAV-mediated gene therapy for muscular dystrophies [319]. The first clinical study was a randomized, double-blind, placebo-controlled phase I clinical trial, conducted in 2012 by Bowles D et al., which used AAV2.5 carrying the mini-dystrophin gene for intramuscular administration in DMD patients [320]. As a result, the administered vector was found to be safe and well-tolerated, and transgene DNA was detected in all patients. Similarly, other clinical trials were started in 2017 and 2018 to test AAV-micro-dystrophin efficacy for treatment of DMD [319] (details in Table 2).

### 5.4. AAV Viral Vectors for Gene Therapy of Cardiovascular and Blood Diseases

Cardiovascular diseases make another attractive target for gene therapy, being a leading cause of death globally with their high incidence and mortality rates [128,321]. As mentioned before, AAV serotypes 6, 8, and 9 have been shown to efficiently transduce cardiac cells in different animal models, and accordingly, they have been used for therapeutic purposes in different cardiovascular diseases [126,128,167,168,322].

In the search for an effective gene therapy platform for heat failure (HF), White J et al., published a study in 2011, in which they suggested a novel technique for AAV-mediated myocardial gene therapy using molecular cardiac surgery [128]. Using AAV6 in an ovine model, the suggested approach resulted in a global transgene expression, that was cardiac-tropic and substantially more robust and targeted, compared to that of intramuscular or intracoronary injection. Furthermore, a randomized phase I/II AAV1-based clinical trial for heart failure treatment was conducted in 2013, where the researchers used a sarcoplasmic reticulum calcium ATPase gene (SERCA2a), the product of which was suggested to play a key role in HF pathology [323]. As a result, adverse effects, including death, were found to be highest in the placebo group, and lowest in the high-dose group, with evidence of long-term transgene expression. However, in the low-dose and mid-dose groups, adverse effects were also found to be high but delayed.

Hemophilia, an inherited disease that is caused either by deficiency of blood clotting factor VIII (type A) or IX (type B), is another example of diseases that have been targeted by AAV-mediated gene therapy [168,322]. rAAV6 and 8 expressing the canine gene of FVIII have been shown to restore physiologic levels of the deficient factor in a canine model for three years following intravenous administration, without any toxicity or immune reactions [322]. The same vectors could result in a similar effect in the murine model; however, neutralizing antibodies against cFVIII were detected in the mice sera [322]. As for hemophilia B, intravenous administration of rAAV8 and 9 carrying the IX factor gene in a murine model resulted in a significant increase in transgene expression and therefore in IX factor levels, with a decreased proinflammatory risk [168].

### 5.5. AAV Viral Vectors for Gene Therapy of Liver Diseases

AAV-based gene therapy plays a significant role in liver diseases, as it is, in some cases, an alternative to the only effective therapy, which is liver transplantation [324]. In addition to the previously mentioned diseases affecting the liver, that have been targeted by AAV-base gene therapy, such as CF, AAT deficiency, and hemophilia, there are other liver diseases that have been targeted and continue to be, using AAV8, mainly.

Wilson’s disease (WD) is a rare autosomal recessive disease caused by mutations in the copper transporter gene, *ATP7B*, resulting in copper accumulation mainly in the liver, along with some other organs [325,326]. A recent in vivo study, conducted by Murillo et al., proved the efficacy of an AAV-based gene therapy for WD in a murine model of the disease [327]. The vector used in the study was AAV8 carrying an optimized short version of the *ATP7B* gene (AAV8-mini *ATP7B*) adjusted to the right size, so it could be produced and delivered more efficiently. As a result, intravenous administration of the recombinant vector to 6- and 12-week-old WD mice could restore copper homeostasis, with 20% hepatocyte transduction being sufficient for correction.

Crigler-Najjar syndrome (CNs) type 1 is another autosomal recessive liver disease caused by deficiency of uridine diphosphate glucuronosyltransferase 1A1 (UGT1A1), which leads to severe inherited unconjugated hyperbilirubinemia [328]. There is currently an ongoing clinical trial investigating AAV8-*UGT1A1* as a vector for gene therapy in children with CNs [329]; however, as the preclinical data of the trial suggest a negative effect of liver cells’ proliferation on the efficacy of gene therapy in children, Shi et al., suggested to optimize the approach by starting therapy at a specific age and combining it with an immune suppression regimen [330]. In their in vivo study, Shi et al., found that a stable correction was achieved when AAV8-*UGT1A1* was administered to CNs rats at the 28th postnatal day, coupled with a rapamycin-based immunosuppression regimen delivered intraperitoneally.

### 5.6. AAV Viral Vectors for Gene Therapy of Endocrine Disorders

Being a major health issue for humans all around the world, with relatively high prevalence estimates, endocrine disorders have always been in the focus of research for therapeutic approaches and techniques [331,332]. Accordingly, rapid developments of diagnostic techniques, along with the better understanding of endocrine disorders’ pathophysiology and molecular bases, helped in developing different therapies, such as hormonal replacement therapies, for example [332,333]. However, as these approaches have mostly focused on improving the patient’s quality of life, reducing or reversing symptoms, rather than targeting the underlying defect, the resulting effect has not always been sufficient, which highlighted the need for gene therapy approaches, including AAV-based ones [333].

Type 1 diabetes mellitus (T1DM) is one of the endocrine autoimmune disorders that has been extensively studied as a target of gene therapy [334]. T1DM is characterized by self-destruction of insulin-secreting islet β cells, resulting from a wide variety of causative factors [335]. In non-obese diabetic (NOD) mice, administration of recombinant adeno-associated virus serotype 8 carrying DNA of mouse insulin promoter (dsAAV8-mIP) has been shown to prevent hyperglycemia in a dose-dependent manner [336]. High levels of mouse interleukin 10 (mIL-10) achieved following rAAV2-*IL-10* intramuscular administration also proved to have a positive effect in NOD mice by decreasing autoimmunity, and thereby hyperglycemia [337]. Similarly, negative regulation of the immune response by programmed death ligand 1 (PDL1) was achieved in NOD mice following intraperitoneal delivery of AAV8-*PDL1*, which protected β cells [338]. Currently, there is an ongoing clinical trial investigating AAV8 containing a transgene for the fragment antigen-binding region of anti-vascular endothelial growth factor (anti-VEGF fab), as a vector for gene therapy of diabetic retinopathy, delivered in the suprachoroidal space [339].

Autoimmune polyglandular syndrome type-1 (APS1) is another disorder characterized by multiple endocrine abnormalities, resulting from a monogenic defect in the autoimmune regulator (*AIRE*) gene [340]. Recently, Almaghrabi tested AAV9-*AIRE* as a potential gene therapy for APS1 [341]. As a result, AAV9-*AIRE*, following intra-thymic administration in a murine model, showed high transduction efficiency, along with restoration of *AIRE* expression in the thymus. Subsequently, a significant reduction of serum auto-antibodies was detected in treated mice, with a relatively normal tissue morphology showing no lymphocytic infiltrations.

### 5.7. AAV Viral Vectors for Gene Therapy of Cancer

Despite the variety of therapeutic approaches developed so far for cancer, including chemotherapy, radiotherapy, surgery, and different medications, cancer is still a huge health issue and a leading cause of death worldwide [342,343]. Several AAV-mediated cancer gene therapy approaches have been reported so far, including suicide gene, RNA-interference, and anti-angiogenesis gene therapies [342].

In AAV-mediated suicide gene therapy, an AAV-based system with herpes simplex virus type-1 thymidine kinase and ganciclovir (AAVtk/GCV) has been used. This system was found to exhibit a significant in vitro and in vivo tumor-suppressor efficacy in human head and neck cancer xenografts in a murine model [344,345,346], and murine models of bile duct cancer [347] and bladder carcinoma [348].

As for AAV-mediated RNA-interference cancer gene therapy, an AAV vector is used to provide a stable expression of short hairpin-structured RNA (shRNA) for gene knockdown applications [349]. This approach could induce a strong androgen receptor (*AR*) gene silencing in a murine model of prostate cancer following intravenous administration [350]. It has also been tested in a murine model of hepatocellular carcinoma and was found to induce tumor-specific apoptosis, resulting in a significant inhibition of cancer cell proliferation without toxicity [351].

AAV-mediated anti-angiogenesis cancer gene therapy targets factors that can regulate a cancer cell angiogenesis process. VEGF, being an important factor for angiogenesis, has been targeted in a murine model of breast carcinoma using AAV2 carrying transgene of VEGF Trap protein (a pseudo high-affinity receptor) [352]. The approach resulted in significant suppression of tumor growth and prevention of spontaneous pulmonary metastases. Another factor that has been targeted by this approach is pigment epithelium-derived factor (PEDF), a highly potent angiogenesis inhibitor. In a murine colorectal peritoneal carcinomatosis model, AAV2 carrying the human gene of PEDF was found to induce significant tumor suppression, inhibit metastases, and prolong survival time [353]. Similarly, kallistatin is another anti-angiogenesis factor that has been targeted using an AAV-mediated murine model of hepatocellular carcinoma, and it provided a similar efficacy as AAV-kallistatin-induced potent cancer cell apoptosis [354].

## 6. Challenges and Limitations of AAVs for Gene Therapy AAV-Mediated Applications

Despite the many advantages AAVs provide as viral vectors for gene therapy of various diseases, and the many improvements and molecular modifications they have undergone since their first discovery, AAVs still have some limitations that can result in impeding impacts on their applications in some cases. One example of such major limitations is the limited cloning capacity of AAVs, that is ~4.5 Kb [355,356]. Moreover, AAVs need to converse into double-stranded DNA, and may cause a delay in transgene expression following delivery, compared to double-stranded vectors [357]. Although it can be dose-dependent in some cases, preexisting immunity against some serotypes can also be a limiting factor [358,359].

Preexisting immunity, or an immunity response occurring after the first introduction of a viral vector, can be a major obstacle in the development of gene therapy approaches based on AAV or other viral vectors, and their application in clinical practice. Neutralizing antibodies are often found in the human population with a high prevalence that varies depending on the virus serotype and the geographic location of the population. Moreover, in a number of diseases, multiple administration of an AAV-based drug may be required, which can result in developing an immune response against the viral vector. Neutralizing antibodies can have a significant impact on the efficiency of gene transfer in gene therapy. Consequently, many patients with genetic diseases will not be able to benefit from such gene therapy medications.

Furthermore, although AAVs have not been linked to human pathologies or diseases, wild-type AAVs are widely distributed in the human population, and despite having a low immunogenicity profile, an immune response was observed in humans as a result of gene therapy using AAVs [358]. To date, the impact of such anti-AAV immune response on gene therapy is still not entirely understood, however, its mechanisms are being actively explored. For example, systemic delivery of AAV vectors, while having neutralizing antibodies, has been shown to result in accumulation of vector genomes in lymphoid organs. Whereas the efficiency of liver transduction while having binding antibodies (indicative of a previous infection, but not neutralizing the virus), on the contrary, increases [360]. The role of neutralizing antibodies in activating the complement system has also been studied [361]. Individuals seropositive for AAV have high titers of IgG1, IgG2, and IgG3. It is known that IgG1, IgG2, and IgM highly correlate with the level of neutralizing antibodies, and IgG1 levels increase after gene therapy with AAV [362]. Additionally, IgG3 correlates with the level of T-cell reactivity against AAV vectors [363]. From a clinical point of view, neutralizing antibody titers are of a great significance, as having neutralizing antibodies against a specific AAV serotype highly affects gene therapy effectiveness upon using this serotype. Although methods for detecting neutralizing antibodies are poorly standardized, existing research data indicate that the prevalence of neutralizing antibodies against AAV2 is about 40% [364]. It has been shown that neutralizing antibodies can trigger elimination of the vector by the immune system. At the same time, viral vectors can accumulate in the spleen, not reaching the liver [365]. Therefore, researching mechanisms to establish and develop effective methods for preexisting immunity assessment can help overcome many problems arising in gene therapy applications. Tackling such an urgent problem can expand the possibilities of therapy for many incurable human hereditary diseases [366].

Another challenge is the need to deeply understand the exact mechanisms and pathways of AAV cellular uptake, trafficking, and transduction, which are still unknown for most AAV serotypes [367]. Such understanding can help in addressing and explaining the changes of transduction potency and efficacy that can be found between different species in different in vivo studies, or between treatment efficacy in vivo and in clinical trials [368]. In other words, differences between humans and various animal models in terms of body size, genomic and anatomic specificity, immunogenicity, and disease process should be taken into account, as they can affect the effective translation of preclinical investigations [369].

**Table 1 cells-12-00785-t001:** Characteristics of different AAV serotypes.

AAV Serotype	Primary Receptor	Other/Coreceptors	Post Translational Modifications	Recommended Purification Method	Tropism
**AAV1**	Sialic acid	AAV receptor (AAVR)	-	iodixanol gradient centrifugation, anion-exchange chromatography, and mucin column affinity chromatography	Skeletal muscles [35,43,44,45,46], heart [48,49], glial and ependymal cells in the murine brain [47], endothelial and vascular smooth muscles [50], retina [51]
**AAV2**	HSPG	FGFR1, αVβ5 and α5β1 integrins, HGFR, LR, and CD9	ubiquitination, phosphorylation, SUMOylation, and multiple-site-glycosylation	heparin column affinity chromatography	renal tissue [62,63,64], hepatocytes [65,66], retina [67,68,69], non-mitotic cells of central nervous system (CNS) [70,71], and skeletal muscles [45,71]
**AAV3**	HSPG	FGFR1, LR, and HGFR	acetylation, phosphorylation, and glycosylation	iodixanol gradient centrifugation	human liver cancer cells as well as human and NHP hepatocytes [76,77], murine cochlear inner hair cells [82].
**AAV4**	Sialic acid	-	ubiquitination	ion-exchange chromatography, mucin column affinity chromatography	ependymal cells of mammalian CNS [90], RPE cells of the retina (canine, rodent, and NHP origins) [91], murine kidney, lung, and heart cells [92,99].
**AAV5**	Sialic acid	PDGFR (α and β)	ubiquitination, phosphorylation, SUMOylation, and glycosylation	ion-exchange chromatography, mucin column affinity chromatography	murine: retinal cells [38,51], mainly photoreceptors [106], airway epithelia [98,110], liver cells [92,112], vascular endothelial cells and smooth muscles [50], and neurons (murine and NHP) [107,370]
**AAV6**	Sialic acid and HSPG	EGFR	acetylation	heparin or mucin column affinity chromatography	airway epithelia of murine and canine models [119,120], murine liver cells [92,121], skeletal muscles of murine and canine models [92,122,123,124], cardiomyocytes in murine [92,125], porcine [126], canine [127], and in sheep [128] models.
**AAV7**	-	-	glycosylation, phosphorylation, SUMOylation, and acetylation	-	murine skeletal muscle cells [133], murine and human hepatocytes [92], murine and NHPs CNS [134,135], murine photoreceptor cells [136], murine vascular endothelial cells (limited tropism) [130], murine epicardium cells [49].
**AAV8**	LR	-	phosphorylation, glycosylation, and acetylation	Dual-ion-exchange chromatography, iodixanol gradient centrifugation	murine, canine, and 115 hepatocytes [92,112,141,142,143,144,145,146,147], murine skeletal and cardiac muscles [148], murine pancreatic cells [149,150], murine renal cells [151], and different cells in the murine retina [152,153,154]
**AAV9**	terminal N-linked galactose	putative integrin, LR	glycosylation, ubiquitination, phosphorylation, SUMOylation, and acetylation	Sucrose gradient centrifugation, and ion-exchange chromatography	murine, NHP, and feline neuronal and non-neuronal cells, including astrocytes [27,159,160,161,162], murine and NHP retinal photoreceptors cells [163], murine, NHP, and porcine cardiac muscle tissue [164,165,166,167,168,169], murine hepatocytes, skeletal muscles, and pancreatic cells [81,169], photoreceptor cells [170], renal tubular epithelium cells [171,172,173], Leydig cells in the testicular interstitial tissue [174], and alveolar and nasal epithelia [175,176].
**AAV10**	-	-	glycosylation, ubiquitination, phosphorylation, SUMOylation, and acetylation	iodixanol gradient centrifugation	NHP intestinal cells, hepatocytes, lymph nodes, and less frequently, renal cells and adrenal glands [180], murine small intestine and colon cells [181], retinal cells (including RPE, cells in the ganglion cell layer, several cell types in the inner nuclear layer, photoreceptors, and a highly efficient transduction of horizontal cells) [152], murine liver cells, lung cells [182], renal, and pancreatic cells [183].
**AAV11**	-	-	-	iodixanol gradient centrifugation	NHP intestinal cells, hepatocytes, lymph nodes, and less frequently, renal cells and adrenal glands [180], murine projection neurons and astrocytes [184], and mild tropism to NHP CNS (cerebrum and spinal cord, mainly) [180].
**AAV12**	mannose and mannosamine have been suggested as components of a potential receptor complex		-	AVB Sepharose affinity chromatography	murine salivary glands and muscles [185], murine nasal epithelia (mainly after intranasal administration) [187].
**AAV13**	-	HSPG	-	iodixanol gradient centrifugation, heparin column affinity chromatography	-

**Table 2 cells-12-00785-t002:** Preclinical and clinical studies using AAVs as viral vectors for gene therapy of different diseases.

Gene Therapy Target	Disease **		AAV Viral Vector	Study Type	Outcome	Ref.
CNS	PD		AAV2-*GAD*	phase I clinical trial for advanced PD patients	safe and well-tolerated approach, providing significant improvement in motor function scores up to 12 months after unilateral subthalamic delivery	[229]
				double-blind, controlled, randomized clinical trial for advanced PD patients	safe and well-tolerated approach, along with improved motor function scores following bilateral subthalamic delivery	[230]
			AAV2-*neurturin*	open-label clinical trial for PD patients	suggested feasibility, safety, and good tolerance of the approach	[231]
			rAAV2-*AADC*	clinical trial for moderately advanced PD patients	good tolerance, improvement on motor rating scales, however, accompanied with an increased risk of intracranial hemorrhages and headache	[232]
					good tolerance and a stable expression of the transgene that lasted for the following 4 years, although higher vector doses were suggested for further studies	[233]
	AADC deficiency		rAAV2-*AADC*	open-label, phase I/II trial in children	good tolerance in general, with evidence for potential improvement of motor function	[234]
	SMA		AA9-*SMN*	open-label, phase I clinical trial	significant improvement of the motor function in all 15 patients following single-dose intravenous administration, reflected by their ability to perform different activities, such as unassisted sitting and walking, oral feeding, and speaking, with no reported motor function regression at two-year follow-up. A long-term safety assessment, however, was recommended.	[236]
	LSD	MPS VII	(rAAV-*GUSβ*)	in vivo study, murine model	stable expression of the deficient enzyme upon single administration, that was adequate for phenotype correction in the liver	[238,239]
			(rAAV-*GUSβ*)	in vivo study, neonatal murine model	therapeutic levels of the deficient enzyme in multiple organs, including ones of the CNS, with the gene expression not being affected by rapid growth and differentiation of tissues	[240]
			rAAV9-*GUSβ* and rAAVrh10-*GUSβ*	in vivo study, canine model	significantly high expression levels of GUSβ in the CNS tissues, with the enzyme in brain tissue homogenates showing over 100% normal activity	[221]
		MPS IIIA	AAV9-*Sgsh* (canine sulfamidase gene)	in vivo study, canine model	sustained and widely distributed transgene expression with no toxicity for a duration of 7 years after therapy	[241]
		MLD	AAV1-*ARSA*	in vivo study, murine model	significant elevation of ARSA levels and activity, resulting in reduction of accumulated sulfatides	[244]
			AAV5-*ARSA*	in vivo study, murine model	rapid, abundant, and sustained restoration of ARSA levels in the brain and brainstem up to 15 months after administration, reduction of accumulated sulfatides, and preservation of neurologic function.	[246]
			AAV5-*ARSA*	in vivo study, NHP model	good tolerance, distribution of the transgene in the brain with elevated activity of the deficient enzyme	[370]
			AAV9-*ARSA*	in vivo study, neonatal murine model	global expression of the transgene in the brain and spinal cord, along with muscles and heart, inhibition of sulfatide accumulation, and improvement of neurologic/motor function	[245]
			AAVrh10-*ARSA* AAVrh10-*ARSA* AAVrh10-*ARSA*	in vivo study, murine model	correction of sulfatide accumulation following single administration, with a transduction efficacy higher than that of AAV5 as it transduced both neurons and oligodendrocytes	[243]
				in vivo study, NHP model	good tolerance, neuroinflammation 3 months following the fifth dose but none after the first, detection of transgene expression after the first dose along with increased ARSA activity, and detection of the enzyme in other organs but not in gonads	[371]
				clinical trial for children with asymptomatic or early-stage MLD	significant elevation in ARSA levels in the cerebrospinal fluid (CSF) following intracerebral vector delivery, but no clinical improvement has been noticed compared to the control group.	[253]
		GM2-gangliosidose	rAAV2-*HEXA* + rAAV2-*HEXB*	in vivo study, murine model	delay of disease clinical onset, maintained motor function, good tolerance, stable and abundant levels of the deficient enzyme, resulting in a significant reduction of gangliosides’ storage in the CNS	[250]
			AAVrh8-*HEX*	in vivo study, feline model	prevention or reduction of tremors that is characteristic of improvement in the neurologic function, and a signification elevation of the deficient enzyme levels (HEX), resulting in a significant reduction of gangliosides’ storage in different tissues of the CNS.	[252]
			AAVrh8-*HEXA* + AAVrh8-*HEXB*	in vivo study, ovine model	delay of disease clinical onset and progression, improved neurologic function and clinical biomarkers. However, lifespan was not significantly higher.	[251]
			AAVrh8-*HEXA* + AAVrh8-*HEXB*	first clinical trial for children with TSD	good tolerance and broad distribution of the transgene in the CNS	[248]
	CD		rAAV2-*ASPA*	in vivo study, murine model	increased ASPA activity and, thereby, decreased NAA accumulation and white matter degeneration. However, areas remote from injection site, such as the cerebellum, were not affected	[256]
	GLD		rAAV1-*GALC*	in vivo study, murine model	sustained expression of the deficient enzyme, improved myelination status, and prolonged lifespan. However, both treated and untreated mice died with similar symptoms, suggesting that the used approach should be initiated prior to symptoms’ onset.	[262]
			rAAV2/5-*GALC*	in vivo study, murine model	wide dispersion of GALC transgene across the CNS reaching areas remote from the injection site, reduced loss of oligodendrocytes and Purkinje cells, along with a significant improvement of neuromotor function and a prolonged lifespan of treated mice.	[263]
			AAV9-*GALC* AAVrh10-*GALC* AAVOlig001-*GALC*	in vivo study, murine model	All three serotypes provided wide distribution of the transgene across the CNS and liver, resulting in a significant improvement of myelination, and a prolonged lifespan, with AAV9 being the most effective when combined with bone marrow transplantation.	[265]
			AAVrh10-c*GALC*	in vivo study, canine model	delayed symptoms onset, prolonged lifespan, correction of biochemical defects, and a positive effect on neuropathology in treated animals	[266].
			AAV9-c*GALC*	in vivo study, canine model	increased activity of the deficient enzyme and, therefore, normal levels of its substrate, improved myelination, and decreased inflammation both in the CNS and PNS, which, along with prevention of clinical neurological dysfunction, resulted in a significantly prolonged lifespan of treated dogs, compared to the control group. However, sufficient dosing was found to be critical, as high doses significantly extended the lifespan even for post-symptomatic subjects, and a 5-fold lower dose of the vector resulted in an attenuated form of disease	[267].
	Retinal degeneration and ocular neurovascular diseases		AAV2, 8, and 9	in vivo study, C57BL/6 mice	transduction of retinal cells, as well as efficient transduction of ganglion cell layer by AAV8 and AAV9	[170]
	Inherited photoreceptor diseases		hybrid serotypes AAV2/7 and AAV2/8	in vivo study, C57BL/6 mice	high transduction rates of the murine photoreceptors	[136]
	Retinal blindness caused by LCA		AAV2/6	in vivo study, wild-type 129Sv/Ev (Taconic) mice	efficient transduction of murine cone photoreceptors following subretinal injection	[268]
			AAV2/4- RPE65	clinical trial for patients with LCA	systemic and local good tolerance with no adverse effects, along with improvement of visual function presented by different parameters in different patients, including improvement of visual acuity and color vision, as well as reduction of visual fatigue or photophobia, over a follow-up period between 1 and 3 years	[269]
	Retinal blindness caused by choroideremia		AAV2- REP1	phase I/II clinical trial	improvement of retinal sensitivity in all six patients, following subretinal vector delivery, with two of them having significant increases in visual acuity, supporting further consideration of the tested therapeutic approach.	[272]
	RPE65-mediated inherited retinal dystrophy		AAV2-hRPE65v2	open-label, randomized, controlled phase III trial	good tolerance with no adverse effects, restoration of RPE65 enzymatic activity reflected by significant and sustained improvement in light perception and navigational abilities	[274]
	Leber Hereditary Optic Neuropathy (LHON)		rAAV2-ND4 (gene encoding nicotinamide adenine dinucleotide dehydrogenase subunit IV)	open-label, phase I/II randomized clinical trial	good tolerance, although a mild, intraocular inflammation was detected after vector administration, but it was responsive to treatment and suggested to be overcome in further studies by systemic vector delivery instead of local.	[275]
				open-label, phase I clinical trial	minor adverse events, improvement in visual activity in some but not all patients, no detection of vector DNA in patients’ blood samples	[276]
	RPGR-related X-linked retinitis pigmentosa		AAV8-RPGR	phase I/II clinical trial	no adverse effects other than steroid-responsive subretinal inflammation following administration of higher doses, sustained improvements in visual function in 6/18 patients.	[277]
	X-Linked Retinoschisis		AAV8-RS1 (Retinoschisin gene)	phase I/IIa single-center, open-label, clinical trial	good tolerance in general, although steroid-responsive, dose-related inflammation was observed, and a dose-related increase of systemic antibodies against AAV8, but none against RS1	[278,279]
	CNGA3-linked achromatopsia		AAV8-CNGA3	nonrandomized controlled clinical trial	no substantial safety issues, successful targeting of cone photoreceptors reflected by reported improvement of color vision ability, along with improvements in visual acuity and contrast sensitivity (although a cause–effect relationship was not established)	[280,281]
	Hearing disorders		AAV9	in vivo study, porcine model	persistent expression of the transgene within the mammalian inner ear following intracochlear delivery	[282]
Respiratory organs (airway epithelia)	CF		AAV2-CFTR	in vivo study-rabbits	efficient and stable gene transfer of CFTR into airway epithelium, indicating, as a result, the vector potential to be used for gene therapy	[286]
				phase I clinical trial for CF patients	safety, successful transduction of targeted cells, and a detected function restoration of sinuses.	[288]
				phase II, double-blind, randomized, placebo-controlled clinical trial for CF patients	safety and good tolerance, but no effective clinical treatment of disease was achieved	[289]
				phase I clinical trial for mild CF patients	safety, but no effective clinical treatment of disease was achieved	[290]
	α1AT deficiency		rAAV-hAAT	in vivo study, intravenous delivery into murine model	Serum levels of AAT that are potentially therapeutic	[297,298]
			rAAV2-hAAT and rAAV5-hAAT	in vivo study, intrapleural vs. intramuscular delivery into murine model	AAT lung and serum levels higher than the ones achieved by intravenous delivery, with rAAV5 showing 10-fold higher effectiveness than rAAV2	[299,300]
			rhAAV10-hAAT	in vivo study, intrapleural delivery into murine and NHP models	safety, and persistent expression of the transgene in the chest cavity cells of both models	[303,304]
			rAAV6/2-hAAT	in vivo study, intratracheal delivery into C57/Bl6 mice, and in vitro study, cultures of human airway epithelial cells	lung cells’ transduction, even more efficient than rAAV5 both in vivo and in vitro	[301]
			rAAV8-hAAT	in vivo study, intratracheal delivery into C57/Bl6 mice	lung cells’ transduction superior to that of rAAV5, as it resulted in 6-fold and 2.5-fold higher AAT levels in serum and broncho-alveolar fluid, respectively.	[302]
			rAAV2/8-hAAT	in vivo study, intravenous delivery into C57/Bl6 mice	high transduction rate of hepatocytes	[298,372]
			rAAV1-AAT	in vivo study, intramuscular delivery into C57/Bl6 mice and rabbits	dose-dependent inflammatory infiltrates at injection sites not affecting expression of transgene, along with dose-dependent detection of vector DNA in most animals, both at injection sites and in distal organs	[46,305]
			rAAV2/9-AAT	in vivo study, intratracheal delivery into C57/Bl6 mice	efficient, relatively stable transduction of alveolar and nasal epithelia, that was not affected by high levels of neutralizing antibodies upon following re-administration	[175]
			rAAV6-hAAT	in vivo study, nasal delivery into C57/Bl6 mice, and intratracheal delivery into canine model	therapeutic hAAT concentrations in both studied animal models, with levels being higher in lungs than serum, accompanied with an immune response against the vector capsid in some animals despite being immunosuppressed	[120]
	Allergic asthma		rAAV-IL-4	in vivo study, intratracheal delivery into Balb/cByJ mice	significant inhibition of airway eosinophilia and mucus production along with a reduction in airway hyper-responsiveness and asthma-associated cytokine levels	[306]
	SPB		rAAV6/2-SFTPB	in vivo study, intratracheal delivery into murine model	efficient transduction of airway and alveolar epithelium, good tolerance of administered vector with no adverse effects, rapid and long-term restoration of the deficient SPB protein, along with an improvement of lung function, leading, subsequently, to an extended survival	[307]
Muscles	Degenerative muscle disorders		AAV1-FS344 (follistatin gene)	in vivo study, NHP model	safety and good tolerance of administered vector, promotion of sustained improvement in muscle size and strength	[311]
	DMD		rAAV6-micro-dystrophin	in vivo study, dystrophin/utrophin double-knockout murine model	no serious adverse events, improvement of muscle function along with prolongation of lifespan, explained by sustained restoration of deficient protein (dystrophin) in the respiratory, cardiac, and limb muscles	[312]
			rAAV1-mini-dystrophin	in vivo study, dystrophin/utrophin double knockout murine model	prolongation of lifespan, highly efficient expression of transgene, improvement in muscle histopathology and function, reflected by improved growth and motility, along with prevention of spine and extremities’ deformation	[313]
			AAV2.5-mini-dystrophin	Randomized, double-blind, placebo-controlled phase I clinical trial	safety and good tolerance of the vector, and detection of transgene DNA in all patients following intramuscular administration	[320]
			AAV9-micro-dystrophin	phase I/II open-label, randomized, controlled clinical trial	safety and successful expression of transgene	[319]
			AAV-rh74-micro-dystrophin	phase I/II, open-label, non-randomized clinical trial	safety and successful expression of transgene, along with improvement of motor function	
	LGMD		rAAV8-hδ-SG (human δ-sarcoglycan gene)	in vivo study, murine model	prolongation of lifespan, efficient and sustained transduction of cardiac and skeletal muscles along with improvement in their histopathology and function	[314]
			rAAV1-hα-SG	in vivo study, murine model	efficient and sustained expression of the transgene, histological and functional improvement of musculature reflected by correction of contractile force deficits and stretch sensibility, along with increase of animal activity	[316]
			rAAV-hγ-SG	in vivo study, murine model	significantly efficient transduction of muscle fibers, and general histopathological improvement, that were achieved only upon early intervention	[317]
Cardiovascular system	HF		AAV6-EGFP	in vivo study, ovine model	global expression of the transgene, that was cardiac-tropic and substantially more robust and targeted, compared to that of intramuscular or intracoronary injection.	[128]
			AAV1-SERCA2a	randomized phase I/II clinical trial	highest adverse effects, including death, in the placebo group, and lowest in the high-dose group, with evidence of long-term transgene expression. However, in the low-dose and mid-dose groups, adverse effects were found to be high but delayed.	[323]
	Hemophilia A		rAAV6-cFVIII and rAAV8-cFVIII	in vivo study, murine and canine models	restoration of physiologic levels of the deficient factor in the canine model for three years following intravenous administration, without any toxicity or immune reactions. A similar effect in the murine model was found, however, with detection of neutralizing antibodies against cFVIII in the mice sera	[322]
	Hemophilia B		rAAV8-FIX and rAAV9-FIX	in vivo study, murine model	significant increase in transgene expression and therefore in IX factor levels, with decreased proinflammatory risk following intravenous administration	[168]
Liver	WD		AAV8-mini ATP7B	in vivo study, murine model	restoration of copper homeostasis, with 20% hepatocyte transduction being sufficient for correction	[327]
	CNs		AAV8- UGT1A1	ongoing clinical trial	-	[329]
				in vivo study, murine model	stable correction was achieved when the vector was administered at the 28th postnatal day, coupled with a rapamycin-based immunosuppression regimen delivered intraperitoneally	[330]
Endocrine system	T1DM		dsAAV8-mIP	in vivo study, murine model	prevention of hyperglycemia in a dose-dependent manner	[336]
			rAAV2-*IL-10*	in vivo study, murine model	a positive effect, decreasing autoimmunity, and thereby hyperglycemia	[337]
			AAV8-*PDL1*	in vivo study, murine model	pancreatic β cells’ protection	[338]
			AAV8-anti-VEGF fab	ongoing clinical trial for diabetic retinopathy patients	-	[339]
	APS1		AAV9-*AIRE*	in vivo study, murine model	high transduction efficiency, along with restoration of AIRE expression in the thymus, following intra-thymic administration, and a subsequent significant reduction of serum auto-antibodies, with relatively normal tissue morphology showing no lymphocytic infiltrations.	[341]
Cancer	laryngeal cancer cell line (HEp-2)		AAVtk/GCV system	in vitro and in vivo study, murine model	significant tumor-suppressor efficacy in human head and neck cancer xenografts, and markedly prolonged survival	[344,345,346]
	bile duct cancer		AAVtk/GCV system	in vivo study, murine model	increased anti-tumor effect upon simultaneous administration with 5-fluorouracil, and better survival	[347]
	bladder carcinoma		AAVtk/GCV system	in vivo study, murine model	control of tumor cell growth, a strong anti-tumor efficacy	[348]
	prostate cancer		AAV2-ARHP8	in vivo study, murine model	strong androgen receptor (AR) gene silencing following intravenous administration	[350]
	hepatocellular carcinoma		AAV8-miR-26a	in vivo study, murine model	Induction of tumor-specific apoptosis, resulting in a significant inhibition of cancer cell proliferation without toxicity	[351]
	breast carcinoma		AAV2-*VEGF Trap*	in vivo study, murine model	significant suppression of tumor growth and prevention of spontaneous pulmonary metastases	[352]
	colorectal peritoneal carcinomatosis		AAV2-h*PEDF*	in vivo study, murine model	significant tumor suppression, inhibition of metastases, and prolonged survival time	[353]
	hepatocellular carcinoma		AAV-*kallistatin*	in vivo study, murine model	induced potent cancer cell apoptosis, and prolonged survival time	[354]

** Disease abbreviations: PD: Parkinson’s disease, AADC deficiency: amino acid decarboxylase deficiency, SMA: spinal muscular atrophy, MPS: mucopolysaccharidosis, MLD: metachromatic leukodystrophy, CD: Canavan’s disease, GLD: globoid cell leukodystrophy, LCA: Leber congenital amaurosis, CF: cystic fibrosis, α1AT deficiency: Alpha-1 antitrypsin deficiency, SPB deficiency: surfactant protein B deficiency, DMD: Duchenne muscular dystrophy, LGMD: Limb-Girdle muscular dystrophies, HF: heart failure, WD: Wilson’s disease, CNs: Crigler-Najjar syndrome, T1DM: Type 1 diabetes mellitus, APS1: autoimmune polyglandular syndrome type-1.

## 7. Conclusions

AAVs were discovered over five decades ago and have since represented a potent tool for gene therapy, that still needs to be better understood and developed for broader and larger therapeutic applications. Further optimizations should cover vector design, tropism modifications, and delivery routes. In this article, we have covered different serotypes of adeno-associated viruses and their applications in gene therapy for different diseases in preclinical and clinical studies, with a brief overview of AAVs’ limitations and challenges.

## Figures and Tables

**Figure 1 cells-12-00785-f001:**
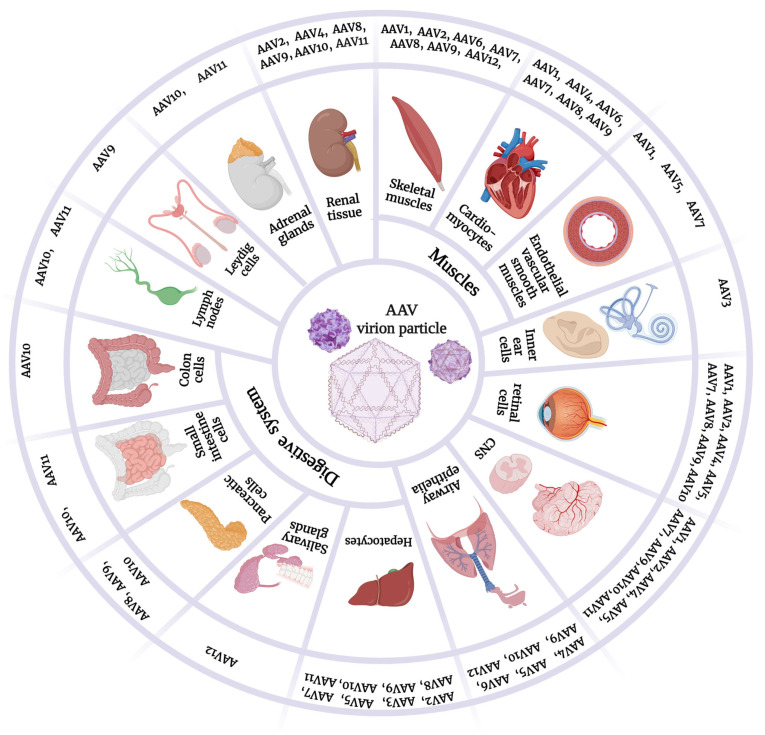
Variant tropisms of AAV serotypes.

## Data Availability

Not applicable.

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
