# Peer review of "Various AAV Serotypes and Their Applications in Gene Therapy: An Overview"

_cells, 2023, doi:10.3390/cells12050785_

Round 1
Reviewer 1 Report
This is a very important and great work! The authors presented a broad literature review describing AAV discovery, properties, different serotypes and tropism, as well as detailed explanation of their pre-clinical and clinical applications in gene therapy for different diseases.
I have no major concerns about this article. My minor comments and recommendations to the authors:
1. Page 13, line 671: please correct the spelling the word “therapy”
”Challenges and limitations of AAVs for gene ther AAV-mediated apy applications”
2. I would advise to authors to add in list of ”AAV viral vectors for gene therapy of CNS” experience of using AAV gene therapy in Krabbe disease and Canavan disease. There are many publications the last few years about laboratory and clinical experience in using AAV gene therapy in these genetic diseases.
Author Response
We would like to thank the editor and the reviewers for their thoughtful comments and efforts towards improving our manuscript. In the following, we highlight our revisions point by point, in order to address all the concerns raised by the first reviewer:
- Typing mistake of the word “therapy”, on page 13, line 671, was corrected.
- Studies for both Canavan’s and Krabbe diseases were added to the section of ”AAV viral vectors for gene therapy of CNS” as suggested by the reviewer.
Reviewer 2 Report
This review does not provide an accurate summary of AAV gene therapy.
Author Response
We thank the reviewer for the valuable comment.Reviewer 3 Report
The manuscript presented for review is a good summary of the knowledge about AAV virus serotypes and their potential use in gene therapy of many diseases. However, the authors should correct the following points:
1. Table No. 1 is a bit illegible, it seems as if the contents of the columns entitled: "PTMs", "Recommended purification method", "Tropism" overlap each other. Maybe to improve the readability of this Table, the pages on which it is located should be in a horizontal layout so that the content of these columns does not overlap
2. In Table 1, the abbreviation PTMs should be expanded, e.g. in the legend to the Table
3. page 17 - the title of Table 2 should start with a capital letter
4. Abbreviations of diseases names from Table No. 2 should be expanded eg in the legend to this Table.
Author Response
We would like to thank the editor and the reviewers for their thoughtful comments and efforts towards improving our manuscript. In the following, we highlight our revisions point by point, in order to address all the concerns raised by the third reviewer:
- All pages with tables were adjusted to landscape layout so they would be readable, as suggested by the reviewer.
- “PTM” abbreviation was re-written in full, as suggested by the editor, and also mentioned in the text above
- The title of Table 2 on page 17 was also corrected to start with a capital letter
- Abbreviations of diseases names from Table No. 2 are also extended in the text above, and they were also added below the table.